# Some *PPF* Dependent Fixed Point Theorems for Generalized *α-F*-Contractions in Banach Spaces and Applications

**Yeol Je Cho [1,2], Shin Min Kang [3,\*] and Peyman Salimi [4]**

[1]  Department of Mathematics Education, Gyeongsang National University, Jinju 52828, Korea; yjcho@gnu.ac.kr
[2]  School of Mathematical Sciences, University of Electronic Science and Technology of China, Chengdu 611731, China
[3]  Department of Mathematics and the RINS, Gyeongsang National University, Jinju 52828, Korea
[4]  Young Researchers and Elite Club, Rasht Branch, Islamic Azad University, P.O. Box 3516-41335, Rasht, Iran; salimipeyman@gmail.com
**\***  Correspondence: smkang@gnu.ac.kr

**Abstract:** In this paper, we introduce the concepts of an *α*-admissible nonself-mapping, an *α-F*-contractive nonself-mapping, a weak *α-F*-contractive nonself-mapping, and a generalized *α-F*-contractive nonself-mapping and prove some *PPF* (past-present-future)-dependent fixed point theorems for the proposed contractive nonself-mappings in certain Razumikhin classes. By using our results, we derive some *PPF*-dependent fixed point theorems for an *α-F*-contractive nonself-mapping endowed with a graph or a partial order. Finally, we give some applications to illustrate the main results.

**Keywords:** Razumikhin class; *PPF*-dependent fixed point; *α*-admissible nonself-mapping; *α-F*-contractive nonself-mapping; weak *α-F*-contractive nonself-mapping; generalized *α-F*-contractive nonself-mapping; graph; (partial) order

## 1. Introduction

Throughout this paper, let $\mathbb{N}$ denote the set of natural numbers, and in addition, for each $h \in \mathbb{N}$, let $\mathbb{N}_h = \{n \in \mathbb{N} : h \le n\}$. Furthermore, let $\mathbb{R}$ and $\mathbb{R}_+ = [0, \infty)$ be the set of all real numbers and the set of all non-negative real numbers, respectively, and let $\mathbb{R}_+^0 = (0, \infty)$.

In 1922, Banach [1] proved that every contraction $T$ in a complete metric space $(X, d)$ has a unique fixed point $z \in X$. Since then, many authors improved and extended this result in several ways (for more details, see [2–17] and the references therein).

Especially, in [18], Bernfeld et al. introduced first the concept of *PPF*-dependent fixed point or a fixed point with *PPF* dependence for nonself-mappings (whose domain is distinct from their range), and furthermore, they showed the existence of *PPF* (past-present-future)-dependent fixed points of Banach-type contraction mappings in certain Razumikhin classes (also, see [19]). Their main results are helpful tools to show the existence of solutions for nonlinear functional differential and integral equations, which may depend on the past history, present data and future evolution (also, see [20]).

On the other hand, in [21], Samet et al. introduced first the concept of *α*-admissible self-mapping, proved some fixed point results for *α*-admissible contractions in a complete metric space, and applied the main results to show the existence of solutions for ordinary differential equations. More recently, in [22], Salimi et al. modified slightly the notions of *α-ψ*-contractive and *α*-admissible mappings and established some fixed point theorems to generalize the results given in [21].

Recently, in [23], Wardowski introduced a new contractive mapping and proved some fixed point theorem for this contraction in complete metric spaces.

For any $k \in (0,1)$, let $\Delta_k$ denote the set of all functions $F : \mathbb{R}^0_+ \to \mathbb{R}$ satisfying the following conditions:

(W1)    $F$ is strictly increasing;

(W2)    for any sequence $(\alpha_n)$ in $\mathbb{R}^0_+$, we have:

$$\lim_{n\to\infty} \alpha_n = 0 \iff \lim_{n\to\infty} F(\alpha_n) = -\infty;$$

(W3)    $\lim_{\alpha \to 0^+} \alpha^k F(\alpha) = 0$.

Denote also $\Delta = \cup\{\Delta_k : k \in (0,1)\}$. An element $F$ in the class $\Delta$ is called a Wardowski function.

Now, assume that $(X, d)$ is a metric space, $F \in \Delta$ and $\tau > 0$. A self-mapping $T : X \to X$ is called an $(F, \tau)$-contractive mapping or $F$-contraction: provided

$$d(Tx, Ty) > 0 \quad \implies \quad \tau + F(d(Tx, Ty)) \leq F(d(x, y)) \tag{1}$$

for all $x, y \in X$.

If $\tau > 0$ is generic in this convention, then $T$ is referred to as an $F$-contraction. A basic example in this direction is as below.

Define a function $F : \mathbb{R}^0_+ \to \mathbb{R}$ by $F(\alpha) = \ln \alpha$ for all $\alpha \in \mathbb{R}^0_+$. Clearly, $F$ satisfies (W1)–(W3). Assume that a mapping $T : X \to X$ is $(F, \tau)$-contractive (in the sense of (1)), where $\tau > 0$, then we have:

$$d(Tx, Ty) \leq e^{-\tau} d(x, y)$$

for all $x, y \in X$ with $Tx \neq Ty$. On the other hand, for all $x, y \in X$ with $Tx = Ty$, we have:

$$d(Tx, Ty) \leq e^{-\tau} d(x, y),$$

which implies that $T$ is the Banach contraction.

In this paper, motivated by the results of Bernfeld et al. [18] and Samet et al. [21,22], we newly introduce the concepts of an $\alpha$-admissible nonself-mapping, an $\alpha$-$F$-contractive nonself-mapping, a weak $\alpha$-$F$-contractive nonself-mapping, and a generalized $\alpha$-$F$-contractive nonself-mapping, and prove some $PPF$-dependent fixed theorems for these kinds of contractive nonself-mappings in Razumikhin classes. By using our results, we prove some $PPF$-dependent fixed point theorems for an $\alpha$-$F$-contractive nonself-mapping when the range space is endowed with the graph or the partial order. Finally, we give some applications to illustrate the main results.

## 2. Preliminaries

Throughout this paper, let $(E, \|\cdot\|_E)$ be a Banach space, $I = [a, b]$ be a closed interval in $\mathbb{R}$, and $E_0 = C(I, E)$ be the set of all continuous $E$-valued functions on $I$ equipped with the supremum norm $\|\cdot\|_{E_0}$ defined as follows:

$$\|\phi\|_{E_0} = \sup_{t \in I} \|\phi(t)\|_E.$$

Let $c \in I$ and $T : E_0 \to E$ be a nonself-mapping.

**Definition 1** ([18]). *We say that a function $\phi \in E_0$ is a PPF-dependent fixed point or a fixed point with PPF dependence of $T$ if $T\phi = \phi(c)$ for some $c \in I$.*

For an easy reference, we list the basic conditions to be used throughout this exposition.

**Definition 2.** *Let $\alpha : E \times E \to [-\infty, +\infty)$ be a nonself-mapping. The mapping $T$ is said to be $\alpha$-admissible if, for any $\phi, \xi \in E_0$,*

$$\alpha\big(\phi(c), \xi(c)\big) \geq 0 \implies \alpha\big(T\phi, T\xi\big) \geq 0.$$

**Example 1.** *Take $E = \mathbb{R}$ (with the usual norm), and let $I = [0, 1]$ and $c = 1$. Define a mapping $T : E_0 \to E$ by $T\phi = \frac{1}{9}\phi(1)$ for all $\phi \in E_0$ and a mapping $\alpha : E \times E \to [-\infty, +\infty)$ by:*

$$\alpha(x, y) = \begin{cases} 2, & \text{if } x \geq y, \\ 0, & \text{otherwise.} \end{cases}$$

*Then, $T$ is an $\alpha$-admissible nonself-mapping.*

**Definition 3.** *Let $\alpha : E \times E \to [-\infty, +\infty)$ be a nonself-mapping and $F \in \Delta$ be a Wardowski function. The mapping $T$ is called:*

*(1)    $\alpha$-F-contractive if there exists $\tau > 0$ such that, for all $\phi, \xi \in E_0$ with $\|T\phi - T\xi\|_E > 0$,*

$$\tau + \alpha\big(\phi(c), \xi(c)\big) + F(\|T\phi - T\xi\|_E) \leq F(\|\phi - \xi\|_{E_0});$$

*(2)    weak $\alpha$-F-contractive if there exists $\tau > 0$ such that, for all $\phi, \xi \in E_0$ with $\|T\phi - T\xi\|_E > 0$,*

$$\tau + \alpha\big(\phi(c), \xi(c)\big) + F(\|T\phi - T\xi\|_E)$$
$$\leq F(\max\{\|\phi - \xi\|_{E_0}, \|\phi(c) - T\phi\|_E, \|\xi(c) - T\xi\|_E\});$$

*(3)    generalized $\alpha$-F-contractive if there exists $\tau > 0$ such that, for all $\phi, \xi \in E_0$ with $\|T\phi - T\xi\|_E > 0$,*

$$\tau + \alpha\big(\phi(c), \xi(c)\big) + F(\|T\phi - T\xi\|_E)$$
$$\leq F\Big( \max \Big\{ \|\phi - \xi\|_{E_0}, \|\phi(c) - T\phi\|_E, \|\xi(c) - T\xi\|_E, \frac{\|\phi(c) - T\xi\|_E + \|\xi(c) - T\phi\|_E}{2} \Big\}\Big).$$

**Definition 4.** *(1)    The Razumikhin (or minimal) class attached to $c$ is defined as follows:*

$$\mathcal{R}_c = \{\phi \in E_0 : \|\phi\|_{E_0} = \|\phi(c)\|_E\}.$$

*(2)    Furthermore, we denote $\mathcal{R}_c^0$ as the class of all constant functions $\phi \in \mathcal{R}_c$, which is referred to as the constant Razumikhin class.*

To get some useful formula for this subclass $\mathcal{R}_c^0$, we need the following:
For each $u \in E$, let $H[u]$ denote the constant function of $E_0$ defined by:

$$H[u](t) = u$$

for all $t \in I$. Note that, by this definition, we have:

$$||H[u]||_{E_0} = ||u||_E, \quad H[u](c) = u$$

whence $H[u] \in \mathcal{R}_c$.

Now, we claim that:

$$\mathcal{R}_c^0 = \{H[u] : u \in E\}.$$

That is, the constant Razumikhin class $\mathcal{R}_c^0$ is the subclass of all constant functions in $E_0$. In fact, the right to left inclusion is clear. For the converse inclusion, it is easy to show that any constant function $\psi$ in $\mathcal{R}_c$ may be written as:

$$\psi = H[u]$$

for some $u \in E$, and so, this completes our argument.

The following simple properties (given without proof) are valid.

**Proposition 5.** *Under the above assumptions,*

*(1)*　　$H[u + v] = H[u] + H[v]$ *for all* $u, v \in E$;
*(2)*　　$H[\lambda u] = \lambda H[u]$ *for all* $\lambda \in \mathbb{R}$ *and* $u \in E$;
*(3)*　　$||u||_E = ||H[u]||_{E_0}$ *for all* $u \in E$;
*(4)*　　*the mapping* $u \mapsto H[u]$ *is an algebraic, topological isomorphism between* $(E, || \cdot ||_E)$ *and* $(\mathcal{R}_c^0, || \cdot ||_{E_0})$.

**Definition 6.** *Let* $\alpha : E \times E \to [-\infty, +\infty)$ *be a nonself-mapping.*

*(1)*　　*the mapping T is said to be* $(\mathcal{R}_c, \alpha)$-*starting if there exists* $\phi_0 \in \mathcal{R}_c$ *such that:*

$$\alpha(\phi_0(c), T\phi_0) \geq 0;$$

*(2)*　　*the mapping T is said to be* $(\mathcal{R}_c^0, \alpha)$-*starting if there exists* $\phi_0 \in \mathcal{R}_c^0$ *such that:*

$$\alpha(\phi_0(c), T\phi_0) \geq 0.$$

Clearly, if $T$ is $(\mathcal{R}_c^0, \alpha)$-starting, then it is also $(\mathcal{R}_c, \alpha)$-starting. The reciprocal assertion is also true under certain regularity conditions upon $T$. Precisely, we have the following:

**Proposition 7.** *Let* $\alpha : E \times E \to [-\infty, +\infty)$ *be a nonself-mapping satisfying the following conditions:*

*(i)*　　*T is* $\alpha$-*admissible;*
*(ii)*　　*T is* $(\mathcal{R}_c, \alpha)$-*starting.*

*Then, T is* $(\mathcal{R}_c^0, \alpha)$-*starting.*

**Proof.** By the second condition above, there exists $\phi_0 \in \mathcal{R}_c$ such that:

$$\alpha(\phi_0(c), T\phi_0) \geq 0.$$

Since $T\phi_0 \in E$, we can consider the element $\xi_0 = H[T\phi_0]$ from the constant Razumikhin class $\mathcal{R}_c^0$, which, by the definition, means that $\xi_0(t) = T\phi_0$ for all $t \in I$, and hence, $\xi_0(c) = T\phi_0$. From the condition on $\phi_0$, it follows that:

$$\alpha(\phi_0(c), \xi_0(c)) \geq 0.$$

Since $T$ is $\alpha$-admissible, we have:

$$\alpha(T\phi_0, T\xi_0) \geq 0$$

or, equivalently, by the preceding relation,

$$\alpha(\xi_0(c), T\xi_0) \geq 0.$$

This completes the proof.　□

**Definition 8.** *(1)*　　*The class* $\mathcal{R}_c$ *is said to be algebraically closed with respect to the difference if* $\phi - \xi \in \mathcal{R}_c$ *for any* $\phi, \xi \in \mathcal{R}_c$;
*(2)*　　*The class* $\mathcal{R}_c$ *is said to be topologically closed if it is closed with respect to the topology on* $E_0$ *generated by the norm* $|| \cdot ||_{E_0}$.

Now, in this paper, we consider a natural continuation of some *PPF*-dependent fixed point theorems proven by Agarwal et al. [24], Ćirić et al. [25], and Hussain et al. [10]. In all the results, they used the following basic structural conditions on the Razumikhin class $\mathcal{R}_c$:

(K1)　　$\mathcal{R}_c$ is algebraically closed with respect to the difference;

(K2)    $\mathcal{R}_c$ is topologically closed.

In this paper, we show that, by the conclusions of the above proposition, both of these conditions may be removed in all results to be presented.

## 3. The Main Results

Let $c \in I$ and $T : E_0 \to E$ be a nonself-mapping.

Now, we need the following result for some *PPF*-dependent fixed point theorems in this section:

**Proposition 9.** *Let $\alpha : E \times E \to [-\infty, +\infty)$ be the nonself-mapping and F be the Wardowski function satisfying the following conditions:*

   *(i)*     *T is $\alpha$-admissible;*
  *(ii)*     *T is a generalized $\alpha$-F-contraction;*
 *(iii)*     *there exists $\phi_0 \in \mathcal{R}_c$ such that $\alpha(\phi_0(c), T\phi_0) \geq 0$.*

   *In addition, assume that:*
   *T has no PPF-dependent fixed points in $\mathcal{R}_c^0$, that is $T\phi \neq \phi(c)$ for all $\phi \in \mathcal{R}_c^0$.*
   *Then, there exists a sequence $(\phi_n)$ in $\mathcal{R}_c^0$, $\phi^* \in \mathcal{R}_c^0$ and $h \in \mathbb{N}$ such that:*

 *(c1)*    *$T\phi_n = \phi_{n+1}(c)$ and $\alpha(\phi_n(c), \phi_{n+1}(c)) \geq 0$ for all $n \in \mathbb{N}$;*
 *(c2)*    *$\phi_n \to \phi^*$ as $n \to \infty$;*
 *(c3)*    *$T\phi_n \neq T\phi^*$, and hence, $\phi_n \neq \phi^*$ for all $n \in \mathbb{N}_h$.*

**Proof.** By a previous observation, Condition (iii) may be written as follows:

 (iv)     there exists $\phi_0 \in \mathcal{R}_c^0$ such that $\alpha(\phi_0(c), T\phi_0) \geq 0$.

Let $\phi_0 \in \mathcal{R}_c^0$. Since $T\phi_0 \in E$, we may consider an element $\phi_1 = H[T\phi_0]$ from the constant Razumikhin class $\mathcal{R}_c^0$, which, by the definition, means that:

$$\phi_1(t) = T\phi_0$$

for all $t \in I$, and hence, $\phi_1(c) = T\phi_0$. Further, since $T\phi_1 \in E$, we may consider an element $\phi_2 = H[T\phi_1]$ from the constant Razumikhin class $\mathcal{R}_c^0$, which, by the definition, means that:

$$\phi_2(t) = T\phi_1$$

for all $t \in I$, and hence, $\phi_2(c) = T\phi_1$. If this process may continue indefinitely, then we can get a sequence $(\phi_n)$ in the constant Razumikhin class $\mathcal{R}_c^0$ with:

$$\phi_n(t) = T\phi_{n-1} \tag{2}$$

for all $n \in \mathbb{N}_1$ and $t \in I$, and hence, $\phi_n(c) = T\phi_{n-1}$. By the algebraic-topological properties of the constant Razumikhin class $\mathcal{R}_c^0$, it follows that:

$$\|\phi_{n-1} - \phi_n\|_{E_0} = \|\phi_{n-1}(c) - \phi_n(c)\|_E$$

for each $n \in \mathbb{N}_1$. Since $T$ is an $\alpha$-admissible mapping and $\alpha(\phi_0(c), \phi_1(c)) = \alpha(\phi_0(c), T\phi_0) \geq 0$, we have:

$$\alpha(\phi_1(c), \phi_2(c)) = \alpha(T\phi_0, T\phi_1) \geq 0.$$

Again, since $T$ is $\alpha$-admissible, we have:

$$\alpha(\phi_2(c), \phi_3(c)) \geq 0.$$

By continuing this process, we have:

$$\alpha(\phi_{n-1}(c), \phi_n(c)) \geq 0$$

for all $n \in \mathbb{N}_1$, and so, this proves Conclusion (c1).

By the imposed condition on the nonself-mapping $T$, the fact that there exists $h \in \mathbb{N}$ such that:

$$T\phi_{h+1} = \phi_{h+1}(c) = T\phi_h$$

is impossible, and so $T\phi_n \neq T\phi_{n+1}$; hence, $\phi_n \neq \phi_{n+1}$ for each $n \in \mathbb{N}$. Since $T$ is a generalized $\alpha$-$F$-contraction, for each $n \in \mathbb{N}$, it follows that:

$$
\begin{aligned}
&\tau + F(\|\phi_n - \phi_{n+1}\|_{E_0}) \\
\leq\ & \tau + \alpha(\phi_{n-1}(c), \phi_n(c)) + F(\|\phi_n - \phi_{n+1}\|_{E_0}) \\
=\ & \tau + \alpha(\phi_{n-1}(c), \phi_n(c)) + F(\|\phi_n(c) - \phi_{n+1}(c)\|_E) \\
=\ & \tau + \alpha(\phi_{n-1}(c), \phi_n(c)) + F(\|T\phi_{n-1} - T\phi_n\|_E) \\
\leq\ & F\Big( \max\Big\{ \|\phi_{n-1} - \phi_n\|_{E_0}, \|\phi_{n-1}(c) - T\phi_{n-1}\|_E, \|\phi_n(c) - T\phi_n\|_E, \\
& \qquad \frac{\|\phi_{n-1}(c) - T\phi_n\|_E + \|\phi_n(c) - T\phi_{n-1}\|_E}{2} \Big\} \Big) \\
=\ & F\Big( \max\Big\{ \|\phi_{n-1} - \phi_n\|_{E_0}, \|\phi_{n-1}(c) - \phi_n(c)\|_E, \|\phi_n(c) - \phi_{n+1}(c)\|_E, \\
& \qquad \frac{\|\phi_{n-1}(c) - \phi_{n+1}(c)\|_E}{2} \Big\} \Big) \\
=\ & F\Big( \max\Big\{ \|\phi_{n-1} - \phi_n\|_{E_0}, \|\phi_{n-1} - \phi_n\|_{E_0}, \|\phi_n - \phi_{n+1}\|_{E_0}, \\
& \qquad \frac{\|\phi_{n-1} - \phi_{n+1}\|_{E_0}}{2} \Big\} \Big) \\
=\ & F\Big( \max\Big\{ \|\phi_{n-1} - \phi_n\|_{E_0}, \|\phi_n - \phi_{n+1}\|_{E_0}, \frac{\|\phi_{n-1} - \phi_{n+1}\|_{E_0}}{2} \Big\} \Big) \\
\leq\ & F\Big( \max\Big\{ \|\phi_{n-1} - \phi_n\|_{E_0}, \|\phi_n - \phi_{n+1}\|_{E_0}, \\
& \qquad \frac{\|\phi_{n-1} - \phi_n\|_{E_0} + \|\phi_n - \phi_{n+1}\|_{E_0}}{2} \Big\} \Big) \\
=\ & F(\max\{\|\phi_{n-1} - \phi_n\|_{E_0}, \|\phi_n - \phi_{n+1}\|_{E_0}\}),
\end{aligned}
$$

which implies that, for each $n \in \mathbb{N}_1$,

$$F(\|\phi_n - \phi_{n+1}\|_{E_0}) \leq F(\max\{\|\phi_{n-1} - \phi_n\|_{E_0}, \|\phi_n - \phi_{n+1}\|_{E_0}\}) - \tau.$$

Now, if $\max\{\|\phi_{n-1} - \phi_n\|_{E_0}, \|\phi_n - \phi_{n+1}\|_{E_0}\} = \|\phi_n - \phi_{n+1}\|_{E_0}$, then we have:

$$F(\|\phi_n - \phi_{n+1}\|_{E_0}) \leq F(\|\phi_n - \phi_{n+1}\|_{E_0}) - \tau < F(\|\phi_n - \phi_{n+1}\|_{E_0}),$$

which is a contradiction. Therefore, we have:

$$F(\|\phi_n - \phi_{n+1}\|_{E_0}) \leq F(\|\phi_{n-1} - \phi_n\|_{E_0}) - \tau$$

and so, for each $n \in \mathbb{N}$,

$$
\begin{aligned}
F(\|\phi_n - \phi_{n+1}\|_{E_0}) \ & \leq F(\|\phi_{n-1} - \phi_n\|_{E_0}) - \tau \\
& \leq F(\|\phi_{n-2} - \phi_{n-1}\|_{E_0}) - 2\tau \\
& \leq \cdots \\
& \leq F(\|\phi_0 - \phi_1\|_{E_0}) - n\tau.
\end{aligned}
\tag{3}
$$

Hence, $\lim_{n\to\infty} F(\|\phi_n - \phi_{n+1}\|_{E_0}) = -\infty$. Since $F \in \Delta$, it follows that:

$$\lim_{n\to\infty} \|\phi_n - \phi_{n+1}\|_{E_0} = 0.$$

Again, since $F \in \Delta$, there exists $k \in (0, 1)$ such that:

$$\lim_{n\to\infty} \|\phi_n - \phi_{n+1}\|_{E_0}^k F(\|\phi_n - \phi_{n+1}\|_{E_0}) = 0.$$

From (3), it follows that, for all $n \in \mathbb{N}$,

$$0 \le n\tau\|\phi_n - \phi_{n+1}\|_{E_0}^k \le \|\phi_n - \phi_{n+1}\|_{E_0}^k \left[ F(\|\phi_0 - \phi_1\|_{E_0}) - F(\|\phi_n - \phi_{n+1}\|_{E_0}) \right].$$

Taking $n \to \infty$, we obtain:

$$\lim_{n\to\infty} n\|\phi_n - \phi_{n+1}\|_{E_0}^k = 0.$$

Therefore, there exists $j \in \mathbb{N}$ such that:

$$n\|\phi_n - \phi_{n+1}\|_{E_0}^k \le 1$$

for all $n \in \mathbb{N}_j$. This implies:

$$\|\phi_n - \phi_{n+1}\|_{E_0} \le \frac{1}{n^{1/k}}$$

for each $n \in \mathbb{N}_j$. Thus, for each $m > n \ge j$, we have:

$$\|\phi_n - \phi_m\|_{E_0} \le \sum_{i=n}^{m-1} \|\phi_i - \phi_{i+1}\|_{E_0} \le \sum_{i=n}^{m-1} \frac{1}{i^{1/k}}.$$

Since $0 < k < 1$, the series $\sum_{i\ge1} \frac{1}{i^{1/k}}$ converges, and so $\|\phi_n - \phi_m\|_{E_0} \to 0$ as $m, n \to \infty$, which implies that $(\phi_n)$ is a Cauchy sequence. Since $\mathcal{R}_c^0$ is complete, there exists $\phi^* \in \mathcal{R}_c^0$ such that $\phi_n \to \phi^*$ as $n \to \infty$. Therefore, (c2) holds too.

Finally, assume that (c3) is not true, that is for each $n \in \mathbb{N}$, there exists $m > n$ such that:

$$T\phi^* = T\phi_m = \phi_{m+1}(c),$$

which implies that there exists an infinite sequence $(k(n))$ in $N$ such that:

$$T\phi^* = \phi_{k(n)}(c)$$

for each $n \in \mathbb{N}$. Taking $n \to \infty$, we have $T\phi^* = \phi^*(c)$, which contracts the imposed hypothesis. Hence conclusion (c3) holds. This completes the proof. $\square$

By using Proposition 9, we have the following basic *PPF*-dependent fixed point theorems:

**Theorem 10.** *Let $\alpha : E \times E \to [-\infty, +\infty)$ be a nonself-mapping and $F$ be a Wardowski function satisfying the following conditions:*

- (i)    *$T$ is $\alpha$-admissible;*
- (ii)   *$T$ is an $\alpha$-$F$-contraction;*
- (iii)  *if $(\phi_n)$ is a sequence in $E_0$ such that $\phi_n \to \phi$ as $n \to \infty$ and $\alpha(\phi_n(c), \phi_{n+1}(c)) \ge 0$ for each $n \in \mathbb{N}$, then $\alpha(\phi_n(c), \phi(c)) \ge 0$ for each $n \in \mathbb{N}$;*
- (iv)   *there exists $\phi_0 \in \mathcal{R}_c$ such that $\alpha(\phi_0(c), T\phi_0) \ge 0$.*

*Then, $T$ has a PPF-dependent fixed point in $\mathcal{R}_c^0$.*

**Proof.** Assume that the conclusion is not true. Since $F$ is strictly increasing, every $\alpha$-$F$-contraction is a generalized $\alpha$-$F$-contraction. Then, all the conditions of Proposition 9 hold. Thus, there exists a sequence $(\phi_n)$ in $\mathcal{R}_c^0$, $\phi^* \in \mathcal{R}_c^0$ and $h \in \mathbb{N}$ such that:

(c1)    $T\phi_n = \phi_{n+1}(c)$ and $\alpha(\phi_n(c), \phi_{n+1}(c)) \geq 0$ for each $n \in \mathbb{N}$;

(c2)    $\phi_n \to \phi^*$ as $n \to \infty$;

(c3)    $T\phi_n \neq T\phi^*$, and hence, $\phi_n \neq \phi^*$ for each $n \in \mathbb{N}_h$.

Since $T$ is an $\alpha$-$F$-contraction, for each $n \in \mathbb{N}_h$, we have:

$$F(\|T\phi_n - T\phi^*\|_E) \leq \tau + \alpha(\phi_n(c), \phi^*(c)) + F(\|T\phi_n - T\phi^*\|_E) \leq F(\|\phi_n - \phi^*\|_{E_0}).$$

Since $F \in \Delta$, for each $n \in \mathbb{N}_h$, we have:

$$\|T\phi_n - T\phi^*\|_E \leq \|\phi_n - \phi^*\|_{E_0}$$

and so:

$$
\begin{aligned}
\|T\phi^* - \phi^*(c)\|_E &\leq \|T\phi^* - T\phi_n\|_E + \|T\phi_n - \phi^*(c)\|_E \\
&= \|T\phi^* - T\phi_n\|_E + \|\phi_{n+1}(c) - \phi^*(c)\|_E \\
&\leq \|\phi^* - \phi_n\|_{E_0} + \|\phi_{n+1} - \phi^*\|_{E_0}.
\end{aligned}
$$

Taking $n \to \infty$, it follows that $\|T\phi^* - \phi^*(c)\|_E = 0$, that is, $T\phi^* = \phi^*(c)$, which is a contradiction. This completes the proof. $\square$

**Theorem 11.** *Let $\alpha : E \times E \to [-\infty, +\infty)$ be a nonself-mapping and $F$ be a Wardowski function satisfying the following conditions:*

(i)    *$T$ is $\alpha$-admissible;*

(ii)    *$T$ is a generalized $\alpha$-$F$-contraction, and $F$ is continuous;*

(iii)    *if $(\phi_n)$ is a sequence in $E_0$ such that $\phi_n \to \phi$ as $n \to \infty$ and $\alpha(\phi_n(c), \phi_{n+1}(c)) \geq 0$ for each $n \in \mathbb{N}$, then $\alpha(\phi_n(c), \phi(c)) \geq 0$ for each $n \in \mathbb{N}$;*

(iv)    *there exists $\phi_0 \in \mathcal{R}_c$ such that $\alpha(\phi_0(c), T\phi_0) \geq 0$.*

*Then, $T$ has a PPF-dependent fixed point in $\mathcal{R}_c^0$.*

**Proof.** Assume that the conclusion of the statement is not true. Since $T$ is a generalized $\alpha$-$F$-contraction, it follows from Proposition 9 that there exists a sequence $(\phi_n)$ in $\mathcal{R}_c^0$, $\phi^* \in \mathcal{R}_c^0$, and $h \in \mathbb{N}$ such that:

(c1)    $T\phi_n = \phi_{n+1}(c)$ and $\alpha(\phi_n(c), \phi_{n+1}(c)) \geq 0$ for each $n \in \mathbb{N}$;

(c2)    $\phi_n \to \phi^*$ as $n \to \infty$;

(c3)    $T\phi_n \neq T\phi^*$, and hence, $\phi_n \neq \phi^*$ for each $n \in \mathbb{N}_h$.

Since $T$ is a generalized $\alpha$-$F$-contraction, for each $n \in \mathbb{N}_h$, we have:

$$
\begin{aligned}
&\tau + F(\|\phi_{n+1}(c) - T\phi^*\|_E) \\
\leq\ & \tau + \alpha(\phi_n(c), \phi^*(c)) + F(\|\phi_{n+1}(c) - T\phi^*\|_E) \\
=\ & \tau + \alpha(\phi_n(c), \phi^*(c)) + F(\|T\phi_n - T\phi^*\|_E) \\
\leq\ & F\Big( \max\Big\{ \|\phi_n - \phi^*\|_{E_0}, \|\phi_n(c) - T\phi_n\|_E, \|\phi^*(c) - T\phi^*\|_E, \\
& \qquad \frac{\|\phi_n(c) - T\phi^*\|_E + \|\phi^*(c) - T\phi_n\|_E}{2} \Big\}\Big) \\
\leq\ & F\Big( \max\Big\{ \|\phi_n - \phi^*\|_{E_0}, \|\phi_n(c) - \phi_{n+1}(c)\|_E, \|\phi^*(c) - T\phi^*\|_E, \\
& \qquad \frac{\|\phi_n(c) - T\phi^*\|_E + \|\phi^*(c) - \phi_{n+1}(c)\|_E}{2} \Big\}\Big).
\end{aligned}
$$

Since $F$ is continuous, by taking $n \to \infty$, it follows that:

$$\tau + F(\|\phi^*(c) - T\phi^*\|_E) \leq F(\|\phi^*(c) - T\phi^*\|_E),$$

which is possible only when $\|\phi^*(c) - T\phi^*\|_E = 0$, that is, $\phi^*(c) = T\phi^*$, which is contradiction. This completes the proof. $\square$

If we use the proof lines of Theorem 11, then we can prove the following:

**Theorem 12.** *Let $\alpha : E \times E \to [-\infty, +\infty)$ be a nonself-mapping and $F$ be a Wardowski function satisfying the following conditions:*

    *(i)    $T$ is $\alpha$-admissible;*
   *(ii)   $T$ is a weak $\alpha$-$F$-contraction, and $F$ is continuous;*
  *(iii)  if $(\phi_n)$ is a sequence in $E_0$ such that $\phi_n \to \phi$ as $n \to \infty$ and $\alpha(\phi_n(c), \phi_{n+1}(c)) \geq 0$ for each $n \in \mathbb{N}$, then $\alpha(\phi_n(c), \phi(c)) \geq 0$ for each $n \in \mathbb{N}$;*
  *(iv)  there exists $\phi_0 \in \mathcal{R}_c$ such that $\alpha(\phi_0(c), T\phi_0) \geq 0$.*

*Then, $T$ has a PPF-dependent fixed point in $\mathcal{R}_c^0$.*

Now, we prove a Suzuki-type theorem for $\alpha$-$F$-contractions in the described Razumikhin class.

**Theorem 13.** *Let $\alpha : E \times E \to [-\infty, +\infty)$ be a nonself-mapping, $F$ be a Wardowski function, and the number $\tau > 0$ be such that:*

    *(i)    $T$ is $\alpha$-admissible;*
   *(ii)   for all $\phi, \xi \in E_0$ with $\|T\phi - T\xi\|_E > 0$ and $\frac{1}{2}\|\phi(c) - T\phi\|_E \leq \|\phi - \xi\|_{E_0}$,*

$$\tau + \alpha(\phi(c), \xi(c)) + F(\|T\phi - T\xi\|_E) \leq F(\|\phi - \xi\|_{E_0}); \tag{4}$$

  *(iii)  if $(\phi_n)$ is a sequence in $E_0$ such that $\phi_n \to \phi$ as $n \to \infty$ and $\alpha(\phi_n(c), \phi_{n+1}(c)) \geq 0$ for each $n \in \mathbb{N}$, then $\alpha(\phi_n(c), \phi(c)) \geq 0$ for each $n \in \mathbb{N}$;*
  *(iv)  there exists $\phi_0 \in \mathcal{R}_c$ such that $\alpha(\phi_0(c), T\phi_0) \geq 0$.*

*Then, $T$ has a PPF-dependent fixed point in $\mathcal{R}_c^0$.*

**Proof.** Assume that the conclusion is not true. By a previous observation, Condition (iv) may be written as follows:

   *(v)   there exists $\phi_0 \in \mathcal{R}_c^0$ such that $\alpha(\phi_0(c), T\phi_0) \geq 0$.*

Let $\phi_0 \in \mathcal{R}_c^0$. Since $T\phi_0 \in E$, we may consider an element $\phi_1 = H[T\phi_0]$ from the constant Razumikhin class $\mathcal{R}_c^0$, which, by the definition, means that:

$$\phi_1(t) = T\phi_0$$

for all $t \in I$ and hence $\phi_1(c) = T\phi_0$. Further, since $T\phi_1 \in E$, we may consider an element $\phi_2 = H[T\phi_1]$ from the constant Razumikhin class $\mathcal{R}_c^0$, which, by the definition, means that

$$\phi_2(t) = T\phi_1$$

for all $t \in I$, and hence, $\phi_2(c) = T\phi_1$. Thus, inductively, we can have a sequence $(\phi_n)$ in the constant Razumikhin class $\mathcal{R}_c^0$ such that, for each $n \in \mathbb{N}_1$,

$$\phi_n(t) = T\phi_{n-1} \tag{5}$$

for all $t \in I$, and hence, $\phi_n(c) = T\phi_{n-1}$. By the algebraic-topological properties of the constant Razumikhin class $\mathcal{R}_c^0$, it follows that:

$$\|\phi_{n-1} - \phi_n\|_{E_0} = \|\phi_{n-1}(c) - \phi_n(c)\|_E$$

for each $n \in \mathbb{N}_1$. Since $T$ is $\alpha$-admissible and:

$$\alpha(\phi_0(c), \phi_1(c)) = \alpha(\phi_0(c), T\phi_0) \geq 0,$$

we have:

$$\alpha(\phi_1(c), \phi_2(c)) = \alpha(T\phi_0, T\phi_1) \geq 0.$$

Again, since $T$ is an $\alpha$-admissible nonself-mapping, we have:

$$\alpha(\phi_2(c), \phi_3(c)) \geq 0.$$

By continuing this process, we have:

$$\alpha(\phi_{n-1}(c), \phi_n(c)) \geq 0$$

for each $n \in \mathbb{N}_1$. By the imposed condition on the nonself-mapping $T$, we cannot have a relation like:

$$T\phi_h = \phi_h(c) = T\phi_{h-1}$$

for some $h \in \mathbb{N}_1$, and so, we have:

$$T\phi_n \neq T\phi_{n+1}$$

and hence, $\phi_n \neq \phi_{n+1}$ for each $n \in \mathbb{N}$.

Now, we have:

$$
\begin{aligned}
\frac{1}{2}\|\phi_{n-1}(c) - T\phi_{n-1}\|_E &= \frac{1}{2}\|\phi_{n-1}(c) - \phi_n(c)\|_E \\
&= \frac{1}{2}\|\phi_{n-1} - \phi_n\|_{E_0} \\
&\leq \|\phi_{n-1} - \phi_n\|_{E_0}.
\end{aligned}
$$

From the Suzuki-type condition we admitted, it follows that:

$$F(\|\phi_n - \phi_{n+1}\|_{E_0}) \leq F(\|\phi_{n-1} - \phi_n\|_{E_0}) - \tau \tag{6}$$

for each $n \in \mathbb{N}_1$. Again, as in the proof of Proposition 9, we can prove that $(\phi_n)$ is a Cauchy sequence in $\mathcal{R}_c^0$, and so, there exists $\phi^* \in \mathcal{R}_c^0$ such that $\phi_n \to \phi^*$ as $n \to \infty$. From (6), we have:

$$F(\|\phi_n - \phi_{n+1}\|_{E_0}) \leq F(\|\phi_{n-1} - \phi_n\|_{E_0}) - \tau < F(\|\phi_{n-1} - \phi_n\|_{E_0})$$

for each $n \in \mathbb{N}_1$. Since $F \in \Delta$, we have:

$$\|\phi_n - \phi_{n+1}\|_{E_0} < \|\phi_{n-1} - \phi_n\|_{E_0} \tag{7}$$

for each $n \in \mathbb{N}_1$. Suppose that there exists $j \in \mathbb{N}$ such that:

$$\frac{1}{2}\|\phi_j(c) - T\phi_j\|_E > \|\phi_j - \phi^*\|_{E_0}$$

and:

$$\frac{1}{2}\|\phi_{j+1}(c) - T\phi_{j+1}\|_E > \|\phi_{j+1} - \phi^*\|_{E_0}.$$

Then, from (7), it follows that:

$$
\begin{aligned}
\|\phi_j - \phi_{j+1}\|_{E_0} &\leq \|\phi_j - \phi^*\|_{E_0} + \|\phi_{j+1} - \phi^*\|_{E_0} \\
&< \frac{1}{2}\|\phi_j(c) - T\phi_j\|_E + \frac{1}{2}\|\phi_{j+1}(c) - T\phi_{j+1}\|_E \\
&= \frac{1}{2}\|\phi_j(c) - \phi_{j+1}(c)\|_E + \frac{1}{2}\|\phi_{j+1}(c) - \phi_{j+2}(c)\|_E \\
&= \frac{1}{2}\|\phi_j - \phi_{j+1}\|_{E_0} + \frac{1}{2}\|\phi_{j+1} - \phi_{j+2}\|_{E_0} \\
&< \frac{1}{2}\|\phi_j - \phi_{j+1}\|_{E_0} + \frac{1}{2}\|\phi_j - \phi_{j+1}\|_{E_0} = \|\phi_j - \phi_{j+1}\|_{E_0},
\end{aligned}
$$

which is a contradiction. Hence, for each $n \in \mathbb{N}$, we have either:

$$\frac{1}{2}\|\phi_n(c) - T\phi_n\|_E \leq \|\phi_n - \phi^*\|_{E_0}$$

or:

$$\frac{1}{2}\|\phi_{n+1}(c) - T\phi_{n+1}\|_E \leq \|\phi_{n+1} - \phi^*\|_{E_0}.$$

Consequently, there exists a sequence $(k(n))$ in $N$ such that, for each $n \in \mathbb{N}$,

$$\frac{1}{2}\|\phi_{k(n)}(c) - T\phi_{k(n)}\|_E \leq \|\phi_{k(n)} - \phi^*\|_{E_0}.$$

Then, from (4), it follows that:

$$
\begin{aligned}
F(\|T\phi_{k(n)} - T\phi^*\|_E) &\leq \tau + \alpha\big(\phi_{k(n)}(c), \phi^*(c)\big) + F(\|T\phi_{k(n)} - T\phi^*\|_E) \\
&\leq F(\|\phi_{k(n)} - \phi^*\|_{E_0}).
\end{aligned}
$$

Now, since $F \in \Delta$, for each $n \in \mathbb{N}$, we have:

$$\|T\phi_{k(n)} - T\phi^*\|_E \leq \|\phi_{k(n)} - \phi^*\|_{E_0}$$

or, equivalently,

$$\|\phi_{k(n)+1}(c) - T\phi^*\|_E \leq \|\phi_{k(n)} - \phi^*\|_{E_0}$$

for each $n \in \mathbb{N}$. Taking $n \to \infty$, it follows that $\|T\phi^* - \phi^*(c)\|_E = 0$, that is, $T\phi^* = \phi^*(c)$. This contradiction completes the proof. □

**Corollary 14.** *Let F be the Wardowski function and the number $\tau > 0$ be such that, for all $\phi, \xi \in E_0$ with $\|T\phi - T\xi\|_E > 0$ and $\frac{1}{2}\|\phi(c) - T\phi\|_E \leq \|\phi - \xi\|_{E_0}$,*

$$\tau + F(\|T\phi - T\xi\|_E) \leq F(\|\phi - \xi\|_{E_0}). \tag{8}$$

*Then, we have the following:*

(1)     *T has a PPF-dependent fixed point in $\mathcal{R}_c^0$;*
(2)     *T has a unique PPF-dependent fixed point in $\mathcal{R}_c$.*

**Proof.** Let $\alpha(u, v) = 0$ for all $u, v \in E$ in Theorem 13. Then, we can prove that $T$ has a *PPF*-dependent fixed point in $\mathcal{R}_c^0$.

For the uniqueness of the *PPF*-dependent fixed point of $T$, suppose that $\phi^*$ and $\xi^*$ are two *PPF*-dependent fixed points of $T$ in $\mathcal{R}_c$ such that $\phi^* \neq \xi^*$. Therefore, we have:

$$\frac{1}{2}\|\phi^*(c) - T\phi^*\|_E = 0 \leq \|\phi^* - \xi^*\|_{E_0}$$

and so, from (8),

$$\begin{aligned}
\tau + F(\|\phi^* - \xi^*\|_{E_0}) &= \tau + F(\|\phi^*(c) - \xi^*(c)\|_E) \\
&= \tau + F(\|T\phi^* - T\xi^*\|_E) \\
&\leq F(\|\phi^* - \xi^*\|_{E_0}),
\end{aligned}$$

which is a contradiction. Hence, $\phi^* = \xi^*$. This completes the proof. □

## 4. Particular Cases

Let $(X, d)$ be a metric space. Consider the directed graph $G$ such that the set $V(G)$ of its vertices coincides with $X$ and the set $E(G)$ of its edges contains all loops, i.e., $E(G) \supseteq \mathcal{I}(X)$ (the diagonal of Cartesian product $X \times X$). If we suppose that $G$ has no parallel edges, then we can identify $G$ with the pair $(V(G), E(G))$. Moreover, we may treat $G$ as the weighted graph (see [26], p. 309) by assigning to each edge the distance between its vertices. If $x$ and $y$ are vertices in the graph $G$, by a path in $G$ from $x$ to $y$ of length $m \in \mathbb{N}_1$, we mean any finite sequence $(x_i)_{i=0}^m$ of $m + 1$ vertices such that $x_0 = x$, $x_m = y$ and $(x_{i-1}, x_i) \in E(G)$ for each $i = 1, \cdots, m$. A graph $G$ is said to be connected if there is a path between any two vertices. The graph $G$ is said to be weakly connected if $\tilde{G}$ is connected (for more details, see [5,9,11,27,28]).

**Definition 15** ([11]). *Let $(X, d)$ be a metric space endowed with a graph $G$. A mapping $T : X \to X$ is called a G-contraction if $T$ preserves the edges of $G$, that is,*

$$(x, y) \in E(G) \implies (Tx, Ty) \in E(G)$$

*and $T$ decreases the weights of the edges of $G$ in the following way: there exists $\alpha \in (0, 1)$ such that:*

$$(x, y) \in E(G) \implies d(Tx, Ty) \leq \alpha d(x, y).$$

In this section, by using the results in Section 3, we give some *PPF*-dependent fixed point theorems in Banach spaces with the graph and the partial order.

### 4.1. Some Results in Banach Spaces Endowed with a Graph

Let $E$ be endowed with the graph $G$, $c \in I$ and $T : E_0 \to E$ be a nonself-mapping.

**Definition 16.** *Let $F \in \Delta$ be a Wardowski function.*

*(1)* *$T$ is called a graphic F-contraction if there exists $\tau > 0$ such that, for all $\phi, \xi \in E_0$ with $(\phi(c), \xi(c)) \in E(G)$ and $\|T\phi - T\xi\|_E > 0$,*

$$\tau + F(\|T\phi - T\xi\|_E) \leq F(\|\phi - \xi\|_{E_0});$$

*(2)* *$T$ is called a graphic weak F-contraction if there exists $\tau > 0$ such that, for all $\phi, \xi \in E_0$ with $(\phi(c), \xi(c)) \in E(G)$ and $\|T\phi - T\xi\|_E > 0$,*

$$\tau + F(\|T\phi - T\xi\|_E) \leq F(\max\{\|\phi - \xi\|_{E_0}, \|\phi(c) - T\phi\|_E, \|\xi(c) - T\xi\|_E\});$$

(3)     *T is called a graphic generalized F-contraction if there exists $\tau > 0$ such that, for all $\phi, \xi \in E_0$ with $(\phi(c), \xi(c)) \in E(G)$ and $\|T\phi - T\xi\|_E > 0$,*

$$\tau + F(\|T\phi - T\xi\|_E) \quad \leq \quad F\Big( \max \Big\{ \|\phi - \xi\|_{E_0}, \|\phi(c) - T\phi\|_E, \|\xi(c) - T\xi\|_E,$$
$$\frac{\|\phi(c) - T\xi\|_E + \|\xi(c) - T\phi\|_E}{2} \Big\} \Big).$$

**Theorem 17.** *Suppose that the following conditions are satisfied:*

(i)    *if $(\phi(c), \xi(c)) \in E(G)$, then $(T\phi, T\xi) \in E(G)$;*
(ii)    *T is a graphic F-contraction, for some Wardowski function F;*
(iii)    *if $(\phi_n)$ is a sequence in $E_0$ such that $\phi_n \to \phi$ as $n \to \infty$ and $(\phi_n(c), \phi_{n+1}(c)) \in E(G)$ for each $n \in \mathbb{N}$, then $(\phi_n(c), \phi(c)) \in E(G)$ for each $n \in \mathbb{N}$;*
(iv)    *there exists $\phi_0 \in \mathcal{R}_c$ such that $(\phi_0(c), T\phi_0) \in E(G)$.*

*Then, T has a PPF-dependent fixed point in $\mathcal{R}_c^0$.*

**Proof.** Define a mapping $\alpha : E \times E \to [-\infty, +\infty)$ by:

$$\alpha(x, y) = \begin{cases} 0, & \text{if } (x, y) \in E(G), \\ \\ -\infty, & \text{otherwise.} \end{cases}$$

Assume that $\alpha(\phi(c), \xi(c)) \geq 0$. Then, $(\phi(c), \xi(c)) \in E(G)$. From (i), we have $(T\phi, T\xi) \in E(G)$, that is $\alpha(T\phi, T\xi) \geq 0$. Hence, $T$ is $\alpha$-admissible.

Let $\phi, \xi \in E_0$ be such that $\|T\phi - T\xi\|_E > 0$. If $\alpha(\phi(c), \xi(c)) \geq 0$, then $(\phi(c), \xi(c)) \in E(G)$. From the definition of a graphic $F$-contraction, we have:

$$\tau + \alpha(\phi(c), \xi(c)) + F(\|T\phi - T\xi\|_E) \leq F(\|\phi - \xi\|_{E_0}).$$

Otherwise, $\alpha(\phi(c), \xi(c)) = -\infty$, that is,

$$\tau + \alpha(\phi(c), \xi(c)) + F(\|T\phi - T\xi\|_E) \leq F(\|\phi - \xi\|_{E_0}).$$

Therefore, for all $\phi, \xi \in E_0$ such that $\|T\phi - T\xi\|_E > 0$, we have:

$$\tau + \alpha(\phi(c), \xi(c)) + F(\|T\phi - T\xi\|_E) \leq F(\|\phi - \xi\|_{E_0}),$$

and $T$ is $\alpha$-$F$-contractive.

Let $(\phi_n)$ be a sequence in $E_0$ such that $\phi_n \to \phi$ as $n \to \infty$ and $\alpha(\phi_n(c), \phi_{n+1}(c)) \geq 0$ for each $n \in \mathbb{N}$. Thus, we have $(\phi_n(c), \phi_{n+1}(c)) \in E(G)$ for each $n \in \mathbb{N}$. In this case, from (iii), it follows that $(\phi_n(c), \phi(c)) \in E(G)$ for each $n \in \mathbb{N}$, that is $\alpha(\phi_n(c), \phi(c)) \geq 0$ for each $n \in \mathbb{N}$.

Finally, from (iv), it follows that there exists $\phi_0 \in \mathcal{R}_c$ such that $\alpha(\phi_0(c), T\phi_0) \geq 0$. Thus, all the conditions of Theorem 10 hold, and so, $T$ has a *PPF*-dependent fixed point. This completes the proof. $\square$

Similarly, we can prove the following:

**Theorem 18.** *Suppose that the following conditions are satisfied:*

(i)    *if $(\phi(c), \xi(c)) \in E(G)$, then $(T\phi, T\xi) \in E(G)$;*
(ii)    *T is a graphic weak F-contraction for a continuous Wardowski function F;*
(iii)    *if $(\phi_n)$ is a sequence in $E_0$ such that $\phi_n \to \phi$ as $n \to \infty$ and $(\phi_n(c), \phi_{n+1}(c)) \in E(G)$ for each $n \in \mathbb{N}$, then $(\phi_n(c), \phi(c)) \in E(G)$ for each $n \in \mathbb{N}$;*

*(iv)*    *there exists $\phi_0 \in \mathcal{R}_c$ such that $(\phi_0(c), T\phi_0) \in E(G)$.*

Then, $T$ has a PPF-dependent fixed point in $\mathcal{R}_c^0$.

**Theorem 19.** *Suppose that the following conditions are satisfied:*

*(i)*    *if $(\phi(c), \xi(c)) \in E(G)$, then $(T\phi, T\xi) \in E(G)$;*
*(ii)*    *$T$ is a graphic generalized F-contraction for a continuous Wardowski function F;*
*(iii)*    *if $(\phi_n)$ is a sequence in $E_0$ such that $\phi_n \to \phi$ as $n \to \infty$ and $(\phi_n(c), \phi_{n+1}(c)) \in E(G)$ for each $n \in \mathbb{N}$, then $(\phi_n(c), \phi(c)) \in E(G)$ for each $n \in \mathbb{N}$;*
*(iv)*    *there exists $\phi_0 \in \mathcal{R}_c$ such that $(\phi_0(c), T\phi_0) \in E(G)$.*

Then, $T$ has a PPF-dependent fixed point in $\mathcal{R}_c^0$.

*4.2. Some Results in Banach Spaces Endowed with a Partial Order*

Let $E$ be a Banach space endowed with a partial order $\preceq$ and $c \in I$ and $T : E_0 \to E$ be a nonself-mapping.

**Definition 20.** *Let $F \in \Delta$ be a Wardowski function.*

*(1)*    *$T$ is called an ordered F-contraction if there exists $\tau > 0$ such that, for all $\phi, \xi \in E_0$ with $\phi(c) \preceq \xi(c)$ and $\|T\phi - T\xi\|_E > 0$,*
$$\tau + F(\|T\phi - T\xi\|_E) \leq F(\|\phi - \xi\|_{E_0});$$

*(2)*    *$T$ is called an ordered weak F-contraction if there exists $\tau > 0$ such that, for all $\phi, \xi \in E_0$ with $\phi(c) \preceq \xi(c)$ and $\|T\phi - T\xi\|_E > 0$,*
$$\tau + F(\|T\phi - T\xi\|_E) \leq F(\max\{\|\phi - \xi\|_{E_0}, \|\phi(c) - T\phi\|_E, \|\xi(c) - T\xi\|_E\});$$

*(3)*    *$T$ is an ordered generalized F-contraction if there exists $\tau > 0$ such that, for all $\phi, \xi \in E_0$ with $\phi(c) \preceq \xi(c)$ and $\|T\phi - T\xi\|_E > 0$,*
$$\tau + F(\|T\phi - T\xi\|_E) \leq F\Big(\max\Big\{\|\phi - \xi\|_{E_0}, \|\phi(c) - T\phi\|_E, \|\xi(c) - T\xi\|_E,$$
$$\frac{\|\phi(c) - T\xi\|_E + \|\xi(c) - T\phi\|_E}{2}\Big\}\Big).$$

**Definition 21** ([10]). *We say that $T$ is c-increasing if, for any $\phi, \xi \in E_0$ with $\phi(c) \preceq \xi(c)$, we have $T\phi \preceq T\xi$.*

**Theorem 22.** *Suppose that the following conditions are satisfied:*

*(i)*    *$T$ is c-increasing;*
*(ii)*    *$T$ is an ordered F-contraction for a Wardowski function F;*
*(iii)*    *if $(\phi_n)$ is a sequence in $E_0$ such that $\phi_n \to \phi$ as $n \to \infty$ and $\phi_n(c) \preceq \phi_{n+1}(c)$ for each $n \in \mathbb{N}$, then $\phi_n(c) \preceq \phi(c)$ for each $n \in \mathbb{N}$;*
*(iv)*    *there exists $\phi_0 \in \mathcal{R}_c$ such that $\phi_0(c) \preceq T\phi_0$.*

Then, $T$ has a PPF-dependent fixed point in $\mathcal{R}_c^0$.

**Proof.** Define a mapping $\alpha : E \times E \to [-\infty, +\infty)$ by:
$$\alpha(x, y) = \begin{cases} 0, & \text{if } x \preceq y \\ -\infty, & \text{otherwise.} \end{cases}$$

Assume that $\alpha(\phi(c), \xi(c)) \geq 0$. Then, we have $\phi(c) \preceq \xi(c)$. Since $T$ is $c$-increasing, it follows that $T\phi \preceq T\xi$, that is $\alpha(T\phi, T\xi) \geq 0$. Thus, $T$ is $\alpha$-admissible.

Let $\phi, \xi \in E_0$ be such that $\|T\phi - T\xi\|_E > 0$. If $\alpha(\phi(c), \xi(c)) \geq 0$, then $\phi(c) \preceq \xi(c))$. From the definition of an ordered $F$-contraction, we have:

$$\tau + \alpha(\phi(c), \xi(c)) + F(\|T\phi - T\xi\|_E) \leq F(\|\phi - \xi\|_{E_0}).$$

Otherwise, $\alpha(\phi(c), \xi(c)) = -\infty$, that is,

$$\tau + \alpha(\phi(c), \xi(c)) + F(\|T\phi - T\xi\|_E) \leq F(\|\phi - \xi\|_{E_0}).$$

Therefore, for all $\phi, \xi \in E_0$ with $\|T\phi - T\xi\|_E > 0$, we have:

$$\tau + \alpha(\phi(c), \xi(c)) + F(\|T\phi - T\xi\|_E) \leq F(\|\phi - \xi\|_{E_0})$$

and $T$ is $\alpha$-$F$-contractive.

Let $(\phi_n)$ be a sequence in $E_0$ such that $\phi_n \to \phi$ as $n \to \infty$ and $\alpha(\phi_n(c), \phi_{n+1}(c)) \geq 0$ for each $n \in \mathbb{N}$. Then, $\phi_n(c) \preceq \phi_{n+1}(c)$ for each $n \in \mathbb{N}$. Thus, from (iii), it follows that $\phi_n(c) \preceq \phi(c)$ for each $n \in \mathbb{N}$, that is $\alpha(\phi_n(c), \phi(c)) \geq 0$ for each $n \in \mathbb{N}$.

Finally, from (iv), it follows that there exists $\phi_0 \in \mathcal{R}_c$ such that $\phi_0(c) \preceq T\phi_0$, that is $\alpha(\phi_0(c), T\phi_0) \geq 0$. Thus, all the conditions of Theorem 10 hold, and so, $T$ has a *PPF*-dependent fixed point in $\mathcal{R}_c^0$. This completes the proof. $\square$

Similarly, we can prove the following:

**Theorem 23.** *Suppose that the following conditions are satisfied:*

   (i)     *T is c-increasing;*

   (ii)    *T is an ordered weak F-contraction for a continuous Wardowski function F;*

   (iii)   *if $(\phi_n)$ is a sequence in $E_0$ such that $\phi_n \to \phi$ as $n \to \infty$ and $\phi_n(c) \preceq \phi_{n+1}(c)$ for each $n \in \mathbb{N}$, then $\phi_n(c) \preceq \phi(c)$ for each $n \in \mathbb{N}$;*

   (iv)   *there exists $\phi_0 \in \mathcal{R}_c$ such that $\phi_0(c) \preceq T\phi_0$.*

*Then, T has a PPF-dependent fixed point in $\mathcal{R}_c^0$.*

**Theorem 24.** *Suppose that the following conditions are satisfied:*

   (i)     *T is c-increasing;*

   (ii)    *T is an ordered generalized F-contraction for a continuous Wardowski function F;*

   (iii)   *if $(\phi_n)$ is a sequence in $E_0$ such that $\phi_n \to \phi$ as $n \to \infty$ and $\phi_n(c) \preceq \phi_{n+1}(c)$ for each $n \in \mathbb{N}$, then $\phi_n(c) \preceq \phi(c)$ for each $n \in \mathbb{N}$;*

   (iv)   *there exists $\phi_0 \in \mathcal{R}_c$ such that $\phi_0(c) \preceq T\phi_0$.*

*Then, T has a PPF-dependent fixed point in $\mathcal{R}_c^0$.*

## 5. Some Applications

In this Section, we give an application to integro-functional equations for some of the results in Section 4. It may be compared with some related statements in Kutbi and Sintunavarat [29].

Let $J = [-a, 0]$ be a bounded closed interval, where $a > 0$ is a real number. Further, let $E = \mathcal{C}(J, R)$ stand for the real space of all continuous functions over $J$. It is easy to see that, with respect to the norm:

(D1)    $\|x\|_E = \max\{|x(t)| : t \in J\}$ for all $x \in E$, the couple $(E, \| \cdot \|_E)$ becomes a Banach space. Moreover, the relation "$\leq$" over $E$ is introduced as follows:

(D2)    $x \leq y$ if and only if $x(t) \leq y(t)$ for all $t \in J$ is reflexive, transitive, and antisymmetric, and hence, the relation "$\leq$" is a partial order on $E$.

Finally, denote $I = [0,1]$, and let $\Lambda = \mathcal{C}(J \cup I, R)$ stand for the linear space of all continuous functions from $J \cup I$ to $R$. As before, with respect to the norm:

(D3)    $||\phi||_\Lambda = \max\{|\phi(t)| : t \in J \cup I\}$ for all $\phi \in \Lambda$, the couple $(\Lambda, || \cdot ||_\Lambda)$ becomes a Banach space.

Now, for each $\phi \in \Lambda$, define the family of delayed functions $\phi_t$ in $E$ such that:

(D4)    $\phi_t(\theta) = \phi(t + \theta)$ for all $t \in I$ and $\theta \in J$. Let $f : I \times E \to R$ be a function such that:

(H1)    $(t, x) \mapsto f(t, x)$ is continuous, that is $t_n \to t$ and $x_n \to x$ imply $f(t_n, x_n) \to f(t, x)$. With the aid of this, for some $z \in E$, we may now consider the following integro-functional equation:

$$\phi_0 = z + \int_0^1 f(s, \phi_s)ds. \tag{9}$$

To show the existence and uniqueness result for the Equation (9), we need some more properties as follows:

Let $F : \mathbb{R}_+^0 \to \mathbb{R}$ be a Wardowski function. We say that $F$ is a normal Wardowski function, provided:

(W4)    $F$ is strictly increasing, continuous, and $F(0+) = -\infty$, $F(\infty) = \infty$, and hence, $F(R_+^0) = R$. In this case, $F$ admits an inverse function $F^{-1} : \mathbb{R} \to \mathbb{R}_+^0$ with the properties

(P1)    $F^{-1}$ is strictly increasing, continuous, and $F^{-1}(-\infty) = 0+$, $F^{-1}(\infty) = \infty$.

Hence, $F$ is a topological isomorphism between $R_+^0$ and $R$. Moreover, for each $\tau > 0$, the associated function $K_\tau : R_+^0 \to \mathbb{R}$ is introduced as follows:

(D5)    $K_\tau(t) = F^{-1}(F(t) - \tau)$ for all $t \in \mathbb{R}_+^0$ is well defined and has the following property:

(P2)    $K_\tau$ is strictly increasing, continuous, and $K_\tau(0+) = 0+$, $K_\tau(\infty) = \infty$, and so, $K_\tau$ is a topological isomorphism from $R_+^0$ to itself. Therefore, if we put $K_\tau(0) = 0$, then the extended function $K_\tau : \mathbb{R}_+ \to \mathbb{R}_+$ is a topological isomorphism from $R_+$ to itself.

Now, we give an application for our result in Section 4.

**Theorem 25.** *Suppose that there exists a function $\alpha \in \Lambda$, a normal Wardowski function $F : \mathbb{R}_+^0 \to \mathbb{R}$ and a number $\tau > 0$ such that (H1) holds, as well as:*

(H2)    *$\phi, \xi \in \Lambda$ and $\phi_0 \leq \xi_0$ imply $f(t, \phi_t) \leq f(t, \xi_t)$ for all $t \in I$;*

(H3)    *$\sup\{||\alpha_t||_E : t \in I\} = ||\alpha_0||_E$ and $\alpha_0 \leq z + \int_0^1 f(s, \alpha_s)ds$;*

(H4)    *$|f(t, x) - f(t, y)| \leq K_\tau(||x - y||_E)$ for all $t \in I$ and $x, y \in E$.*

*Then, the integro-functional Equation (9) has exactly one solution in $\Lambda$.*

**Proof.** Define the following linear space:

(D6)    $\widehat{E} = \{\widehat{\phi} := (\phi_t) : t \in I, \phi \in \Lambda\}$.

**(I-1)** If we define the norm as follows:

(D7)    $\widehat{\phi}_{\widehat{E}} = \sup\{||\phi_t||_E : t \in I\}$ for all $\widehat{\phi} \in \widehat{E}$,

then $\widehat{E}$ becomes a Banach space. In fact, let $(\widehat{\phi}^n)_{n \in \mathbb{N}}$ be a Cauchy sequence in $\widehat{E}$, and hence, by the definition above, we have the following:

(i)     $(\phi^n(t))_{n \in \mathbb{N}}$ is a Cauchy sequence in $R$ for all $t \in J \cup I$;
(ii)    $(\phi_t^n)_{n \in \mathbb{N}}$ is a Cauchy sequence in $E$ for all $t \in I$.

By the former conclusion above, one can derive that:

(a)    $\phi(t) = \lim_{n \to \infty} \phi^n(t)$ exists for all $t \in J \cup I$.

This, combined with the latter conclusion, tells us that:

(b)    $\phi \in \Lambda$ and, hence, $\phi_t \in E$ for all $t \in I$;

(c)    $\phi^n \to \widehat{\phi} := (\phi_t; t \in I)$ with respect to the norm $|| \cdot ||_{\widehat{E}}$,

and so, our claim follows.

**(I-2)** The following regularity property of the structure $(\widehat{E}, ||.||_{\widehat{E}})$ is also valid:

(Reg)    for each sequence $(\widehat{\phi}^n)_{n \in \mathbb{N}}$ in $\widehat{E}$, $\widehat{\phi} \in \widehat{E}$ with $\widehat{\phi}^n \to \widehat{\phi}$ (in the norm $|| \cdot ||_{\widehat{E}}$) and $\phi_0^n \leq \phi_0^{n+1}$ for each $n \in \mathbb{N}$, we have $\phi_0^n \leq \phi_0$ for each $n \in \mathbb{N}$.

This is a direct consequence of the following property:

(d)    $\widehat{\phi}^n \to \widehat{\phi}$ (with respect to the norm $|| \cdot ||_{\widehat{E}}$) implies $\phi_0^n \to \phi_0$ (in $E$), and this, in turn, gives $\phi_0^n \leq \phi_0$ for each $n \in \mathbb{N}$.

By using these properties, let us introduce the operator $\mathcal{T} : \widehat{E} \to E$ defined by:

(Oper)    $\mathcal{T}\widehat{\phi} = z + \int_0^1 f(s, \phi_s) ds$ for all $\widehat{\phi} \in \widehat{E}$.

Now, we claim that the conditions of Theorem 22 hold, and then, this will complete the argument.

**(II-1)** First, by (H2), we have:

(Q1)    $\widehat{\phi}, \widehat{\xi} \in \widehat{E}$ and $\phi_0 \leq \xi_0$ imply $f(s, \phi_s) \leq f(s, \xi_s)$ for all $s \in I$, and this gives $\mathcal{T}\widehat{\phi} \leq \mathcal{T}\widehat{\xi}$ in $E$,

which shows that $\mathcal{T}$ is zero-increasing.

**(II-2)** Secondly, by (H3), there exists $\alpha \in \Lambda$ such that:

(i)    $\sup\{||\alpha_t||_E : t \in I\} = ||\alpha_0||_E$;

(ii)    $\alpha_0 \leq z + \int_0^1 f(s, \alpha_s) ds$.

This tells us that the element $\widehat{\alpha} = (\alpha_t; t \in I)$ in $\widehat{E}$ satisfies:

(Q2)    $||\widehat{\alpha}||_{\widehat{E}} = ||\alpha_0||_E$;

(Q3)    $\alpha_0 \leq z + \int_0^1 f(s, \alpha_s) ds = \mathcal{T}\widehat{\alpha}$.

**(II-3)** Finally, we have to verify the contractive condition. Let $F$, $\tau$ and $K_\tau$ be introduced as before. For any $\widehat{\phi}, \widehat{\xi} \in \widehat{E}$ with $\phi_0 \leq \xi_0$ and $\mathcal{T}\widehat{\phi} \neq \mathcal{T}\widehat{\xi}$, we have, by (H4),

$$
\begin{aligned}
||\mathcal{T}\widehat{\phi} - \mathcal{T}\widehat{\xi}||_E &\leq \int_0^1 |f(s, \phi_s) - f(s, \xi_s)| ds \\
&\leq \int_0^1 K_\tau(||\phi_s - \xi_s||_E) ds \\
&\leq \int_0^1 K_\tau(||\widehat{\phi} - \widehat{\xi}||_{\widehat{E}}) ds \\
&= K_\tau(||\widehat{\phi} - \widehat{\xi}||_{\widehat{E}}) \\
&= F^{-1}(F(\widehat{\phi} - \widehat{\xi}||_{\widehat{E}}) - \tau)
\end{aligned}
$$

and so:

$$
\tau + F(||\mathcal{T}\widehat{\phi} - \mathcal{T}\widehat{\xi}||_E) \leq F(||\widehat{\phi} - \widehat{\xi}||_{\widehat{E}})
$$

or, in other words, $\mathcal{T}$ is ordered $F$-contractive. Putting these together, it follows that Theorem 22 is indeed applicable to our data, and so, from this, we can complete our conclusion.

□

## 6. Conclusions

Motivated by the results of Bernfeld et al. [18] and Samet et al. [21,22], we newly introduce the concepts of an *α*-admissible nonself-mapping, an *α-F*-contractive nonself-mapping, a weak *α-F*-contractive nonself-mapping, and a generalized *α-F*-contractive nonself-mapping, and prove some *PPF*-dependent fixed theorems for these kinds of contractive nonself-mappings in Razumikhin classes. By using our results, we derive some *PPF*-dependent fixed point theorems for an *α-F*-contractive nonself-mapping when the range space is endowed with the graph or the partial order. Furthermore, we give some applications to illustrate the main results.

Finally, we expect that our main results will contribute much to the development of *PPF*-dependent fixed point theory and applications.

**Author Contributions:** All the authors have contributed equally to this paper. All the authors have read and approved the final manuscript.

**Funding:** This work was supported by the Gyeongsang National University Fund for Professors on Sabbatical Leave, 2018.

**Acknowledgments:** All the authors wish to express their warm thanks to D. Goeleven, S. Z. Nemeth and M. Turinici for reading an initial draft of the paper and providing many useful remarks.

**Conflicts of Interest:** The authors declare no conflict of interest.

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
