# Peer review of "Some PPF Dependent Fixed Point Theorems for Generalized α-F-Contractions in Banach Spaces and Applications"

_mathematics, doi:10.3390/math6110267_

Round 1
Reviewer 1 Report
In this manuscript the authors present some interesting results concerning
contractive maps in certain Razumikhin class.
The subjest is interesting, and their research original and the theorems they
proof are valuable.
Moreover, they present a final section with application which is illustrative.
And they also give examples of the different points along the text
The manuscript is well organized and well written, and can be followed easily.
The introduction is rather good, givening generic references of previous work,
and also highlighting some which is particularly relevant for the work.
Summarizing, I recommend its publication in Mathematics.
Some minor comments:
1) Although the acronym 'PPF' used without definition or spelling out in the title,
abstract, and other places along the text, in my opinion is should be spelled out
iat some point, for the benefit of the non-specialized reader.
2) The first paragragh is in my opinion irrelevant in a text on mathematics,
although N for the natural numbers and R for the reals should be typed correctly,
i.e. \mathbb{N} and \mathbb{R} with \usepackage{amsmath,amsfonts,amssymb}
in the preamble.
3) This is also true in other places along the text, v.g. line 58
4) In line 58 k belongs to \mathbb{N} or \mathbb{R}?
5) In line 68, the authors state 'by works above' which ones do they exactly mean?
Reviewer 2 Report
The topic of the manuscript is interesting.
However:
Reference [16] is not cited within the text.
The references are not cited in order of appearance in text, and are not written according to the journal's requirements.
The abstract has 201 words, however, the maximum word count for the abstract allowed by the journal is of 200 words.
There is a letter missing from the word "assuptions" in line 95.
Definition 2.1 is missing "for some c from I".
The authors use the range with minus infinite included in it, without giving details about the algebraic and topological structure of the range "[- infinite, infinite)". The use of minus infinite makes definition 2.2. incorrect. It would mean that minus infinite can be higher or equal to zero, which is false.
Some aspects are not cited, such as phrase 2.1.
The text formatting could be improved. The positioning of formulae within the text could be improved, such as in lines 83, 86. Some formulae are highlighted with empty lines before and after them, some are not. In line 234, should be written p.309, instead of pp.309. Line 249 is missing the period "." at the end.
In line 32, I believe it would have been better if "46N40" followed after "46T99".
The authors have not commented about the possibility to generalize the calculations, using d(Tx, Ty, Tz) instead of d(Tx,Ty).
The authors did not explain why they used the symbol G instead of alpha, which is used by other authors.
In example 2.1, line 82, the authors did not explain why they used 1/5 instead of 1/9, which is used by other authors.
The authors did not write any conclusions at the end of the manuscript.
